# *Bradyrhizobium diazoefficiens* USDA110 Nodulation of *Aeschynomene afraspera* Is Associated with Atypical Terminal Bacteroid Differentiation and Suboptimal Symbiotic Efficiency

Quentin Nicoud,[a] Florian Lamouche,[a] Anaïs Chaumeret,[a] Thierry Balliau,[b] Romain Le Bars,[a] Mickaël Bourge,[a] Fabienne Pierre,[a] Florence Guérard,[c] Erika Sallet,[d] Solenn Tuffigo,[a] Olivier Pierre,[a] Yves Dessaux,[a] Françoise Gilard,[c] Bertrand Gakière,[c] Istvan Nagy,[e,f] Attila Kereszt,[e,f] Michel Zivy,[b] Peter Mergaert,[a] Benjamin Gourion,[d] Benoit Alunni[a]

[a]Université Paris-Saclay, CEA, CNRS, Institute for Integrative Biology of the Cell (I2BC), Gif-sur-Yvette, France
[b]PAPPSO, GQE-Le Moulon, INRAE, CNRS, AgroParisTech, Paris-Saclay University, Gif-sur-Yvette, France
[c]SPOmics platform, Institute of Plant Sciences Paris-Saclay (IPS2), CNRS, INRAE, Universities Paris-Saclay, Evry and de Paris, Orsay, France
[d]LIPM, Université de Toulouse, INRAE, CNRS, Castanet-Tolosan, France
[e]Institute of Biochemistry, Hungarian Academy of Sciences, Biological Research Centre, Szeged, Hungary
[f]Seqomics Biotechnology Ltd., Mórahalom, Hungary

Quentin Nicoud and Florian Lamouche are co-first authors. Author order was determined reverse alphabetically.

**ABSTRACT** Legume plants can form root organs called nodules where they house intracellular symbiotic rhizobium bacteria. Within nodule cells, rhizobia differentiate into bacteroids, which fix nitrogen for the benefit of the plant. Depending on the combination of host plants and rhizobial strains, the output of rhizobium-legume interactions varies from nonfixing associations to symbioses that are highly beneficial for the plant. *Bradyrhizobium diazoefficiens* USDA110 was isolated as a soybean symbiont, but it can also establish a functional symbiotic interaction with *Aeschynomene afraspera*. In contrast to soybean, *A. afraspera* triggers terminal bacteroid differentiation, a process involving bacterial cell elongation, polyploidy, and increased membrane permeability, leading to a loss of bacterial viability while plants increase their symbiotic benefit. A combination of plant metabolomics, bacterial proteomics, and transcriptomics along with cytological analyses were used to study the physiology of USDA110 bacteroids in these two host plants. We show that USDA110 establishes a poorly efficient symbiosis with *A. afraspera* despite the full activation of the bacterial symbiotic program. We found molecular signatures of high levels of stress in *A. afraspera* bacteroids, whereas those of terminal bacteroid differentiation were only partially activated. Finally, we show that in *A. afraspera*, USDA110 bacteroids undergo atypical terminal differentiation hallmarked by the disconnection of the canonical features of this process. This study pinpoints how a rhizobium strain can adapt its physiology to a new host and cope with terminal differentiation when it did not coevolve with such a host.

**IMPORTANCE** Legume-rhizobium symbiosis is a major ecological process in the nitrogen cycle, responsible for the main input of fixed nitrogen into the biosphere. The efficiency of this symbiosis relies on the coevolution of the partners. Some, but not all, legume plants optimize their return on investment in the symbiosis by imposing on their microsymbionts a terminal differentiation program that increases their symbiotic efficiency but imposes a high level of stress and drastically reduces their viability. We combined multi-omics with physiological analyses to show that the symbiotic couple formed by *Bradyrhizobium diazoefficiens* USDA110 and *Aeschynomene afraspera*, in which the host and symbiont did not evolve together, is functional but displays a low symbiotic efficiency associated with a disconnection of terminal bacteroid differentiation features.

Address correspondence to Benoit Alunni, benoit.alunni@i2bc.paris-saclay.fr.

**KEYWORDS** cell differentiation, legume-rhizobium symbiosis, metabolomics, nitrogen fixation, proteomics, transcriptomics

Nitrogen availability is a major limitation for plant development in many environments, including agricultural settings. To overcome this problem and thrive on substrates presenting a low nitrogen content, crops are heavily fertilized, causing important environmental damage and financial drawbacks (1, 2). Plants of the legume family acquired the capacity to form symbiotic associations with soil bacteria, the rhizobia, which fix atmospheric nitrogen for the plants' benefit. These symbiotic associations lead to the development of rhizobium-housing root organs called nodules. In these nodules, the rhizobia adopt an intracellular lifestyle and differentiate into bacteroids that convert atmospheric dinitrogen into ammonia and transfer it to the plant. Critical recognition steps occur all along the symbiotic process and define the compatibility of the plant and bacterial partners (3). While the mechanisms involved at the early stages of the symbiosis are well described, those of the later stages are much less clear and might affect not only the ability to interact but also the efficiency of the symbiosis (i.e., the plant benefit).

Nodule-specific cysteine-rich (NCR) antimicrobial peptides produced by legumes of the dalbergioids and the inverted-repeat-lacking clade (IRLC) were proposed to play a crucial role in the control of host-symbiont specificity at the intracellular stage of the symbiosis (4). NCR peptides are targeted to the bacteroids, where they govern bacteroid differentiation (5–9). In these legumes, the differentiation process entails such profound changes that they suppress the bacteroids' capacity to resume growth and is therefore referred to as terminal bacteroid differentiation (TBD). TBD contrasts with bacteroid formation in legumes that lack NCR genes (e.g., soybean), where bacteroids are in a reversible state and can resume growth when released from nodules (10). Specifically, TBD is associated with cell elongation, an increase in the bacteroid DNA content through a cell cycle switch toward endoreduplication (6, 9, 11). Furthermore, increased permeability of the bacteroid envelope also occurs during TBD, most probably due to the interaction of NCR peptides with bacterial membranes (6, 7, 10, 12). Together, these alterations of bacteroid physiology are associated with a strongly decreased viability of the differentiated bacteria, which fail to recover growth when extracted from nodules (6).

While many rhizobia have a narrow host range, some species can nodulate a large array of plant species. One of them, *Bradyrhizobium diazoefficiens* USDA110, can trigger functional nodules without TBD on soybean (*Glycine max*), cowpea (*Vigna unguiculata*), or siratro (*Macroptilium atropurpureum*) (Fig. 1A and B) (13). In addition to these species, USDA110 also induces functional nodules on the TBD-inducing legume *Aeschynomene afraspera* (Fig. 1A and B) (14, 15). However, in *A. afraspera*, USDA110 shows only very limited features that are usually associated with TBD, suggesting that the bacterium might be resistant to the TBD process (16).

Here, we further characterized bacteroid differentiation in the symbiosis between USDA110 and *A. afraspera*. Our observations, supported by whole-nodule metabolome analysis, indicate that USDA110 is poorly matched for nitrogen fixation with *A. afraspera*. To understand better the adaptation of *B. diazoefficiens* physiology to the *G. max* and *A. afraspera* nodule environments, we used a combination of transcriptomics (transcriptome sequencing [RNA-seq]) and shotgun proteomics (liquid chromatography-tandem mass spectrometry [LC-MS/MS]) approaches. Finally, we find that USDA110 undergoes terminal but atypical bacteroid differentiation in *A. afraspera* with reduced cell viability and increased membrane permeability, while cell size and ploidy levels remain unchanged.

## RESULTS

***B. diazoefficiens* USDA110 is poorly matched with *A. afraspera* for nitrogen fixation.** Previous reports indicate that *B. diazoefficiens* USDA110, the model symbiont of soybean, is able to establish a functional nitrogen-fixing symbiosis with *A. afraspera*,

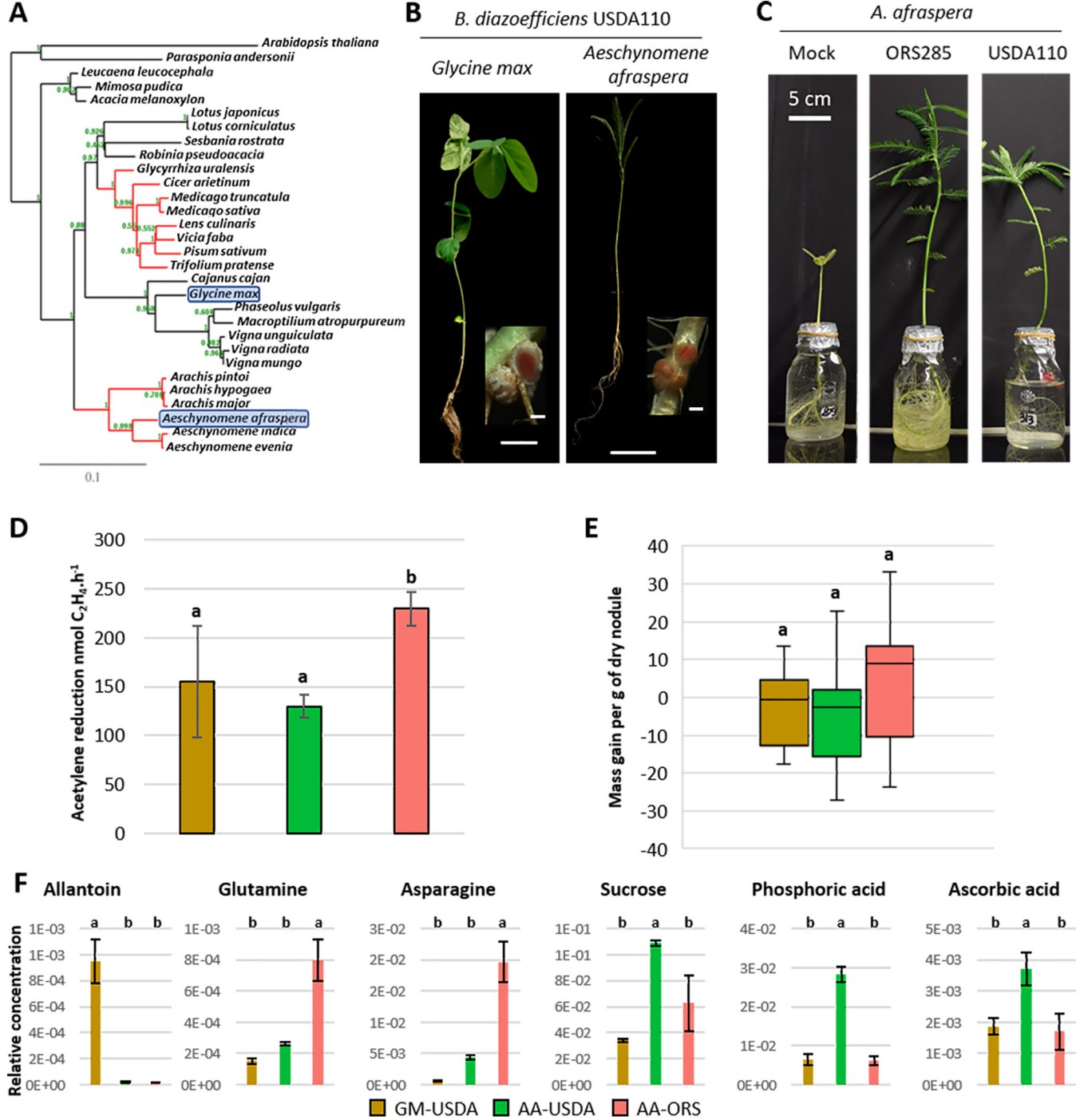

FIG 1 The nonadapted symbiotic couple formed by *Bradyrhizobium diazoefficiens* USDA110 and the NCR-producing plant *Aeschynomene afraspera* displays suboptimal nitrogen fixation and nodule metabolic dysfunction. (A) Phylogenetic ML tree of a selection of plant species based on *matK* nucleotide sequences. Red branches indicate clades of legume plants inducing terminal bacteroid differentiation. Blue boxes indicate the distantly related host plants used in this study. Bootstrap values are mentioned in green on each node of the tree. (B) General aspect of the plants and nodule sections (inlays) displaying the red coloration of leghemoglobin of *G. max* (left) and *A. afraspera* (right) at 14 dpi. Bars, 5 cm (plants) and 1 mm (nodules). (C) Aspect of *A. afraspera* plants nodulated by ORS285, USDA110, or mock-inoculated controls at 21 dpi. (D and E) Nitrogen fixation activity determined by an acetylene reduction assay (D) and gain in biomass attributable to the symbiosis (E) of 14-dpi plants. (F) Whole-nodule metabolome determined by GC-MS or LC-MS at 14 dpi. Histograms show the average values of the relative metabolite concentrations from four biological replicates. Letters represent significant differences after ANOVA and *post hoc* Tukey tests ($P < 0.05$). GM, *G. max* bacteroids; AA, *A. afraspera* bacteroids; USDA, *B. diazoefficiens* USDA110; ORS, *Bradyrhizobium* sp. ORS285.

mSystems®

a phylogenetically distant host belonging to the dalbergioid clade that naturally interacts with photosynthetic rhizobia such as *Bradyrhizobium* sp. strain ORS285 (Fig. 1A to C) (14–18). To evaluate the efficiency of this symbiosis, the nitrogenase activity of USDA110- and ORS285-infected plants and their nitrogen content were determined. Although nitrogenase activity was detected in both types of nodules, it was significantly lower in USDA110-nodulated plants (Fig. 1D). A similar trend was observed for mass gain per nodule mass, although this difference is not significant (Fig. 1E). Nitrogen and carbon contents also seemed to be reduced in USDA110-nodulated plants compared to ORS285-nodulated plants, reaching levels similar to those of noninoculated plants (see Fig. S1 in the supplemental material). Accordingly, ORS285-nodulated *A. afraspera* plants displayed darker green leaves than those interacting with USDA110. Moreover, the plants inoculated with ORS285 are clearly much larger than the ones inoculated with USDA110 at later time points (21 days postinoculation [dpi]) (Fig. 1C), validating our conclusions drawn from the physiological analysis at 14 dpi that USDA110 is a poor symbiotic partner for *A. afraspera*, although the symbiosis is functional.

Moreover, the shoot/root mass ratio, a metric that reflects the nutritional status of the plant, was reduced in USDA110-nodulated *A. afraspera* plants compared to ORS285-nodulated plants, indicating that the plant nutritional needs were not fulfilled (Fig. S2) (19). To characterize further this suboptimal symbiosis, we analyzed the whole-nodule metabolome. Soybean nodules infected with USDA110 were used as a reference (Fig. S3). Allantoin, which is known to be the major nitrogen form exported from soybean nodules, was specifically detected in them (Fig. 1F) (20). On the contrary, asparagine and glutamine are the principal exported nitrogen compounds in *A. afraspera* nodules, and their amount was smaller in USDA110-infected nodules than in ORS285-infected nodules, indicating reduced nitrogen fixation by the bacteroids (Fig. 1F) (18).

In addition, we found specifically in USDA110-infected *A. afraspera* nodules the accumulation of sucrose, phosphoric acid, and ascorbate and, oppositely, a strong reduction in the trehalose content (Fig. 1F; Fig. S3). Sucrose derived from phloem sap is metabolized in nodules to fuel the bacteroids with carbon substrates, usually dicarboxylates. The accumulation of sucrose in nodules indicates symbiotic dysfunction. Also, the accumulation of phosphoric acid in nodules suggests that nitrogen fixation is not reaching its optimal rate (18). Ascorbate has been shown to increase nitrogen fixation activity by modulating the redox status of leghemoglobin (21, 22). Thus, its accumulation in nodules with a reduced nitrogen fixation capacity could be a stress response to rescue nitrogen fixation in nodules that do not fix nitrogen efficiently. A trehalose biosynthesis gene is upregulated in ORS285 bacteroids in *A. afraspera*, suggesting that TBD is accompanied by the synthesis of this osmoprotectant disaccharide (17). The lower synthesis in USDA110 bacteroids suggests an altered TBD. Together, these data indicate metabolic disorder in the USDA110-infected nodules, in agreement with USDA110 being a suboptimal symbiont of *A. afraspera*.

**Overview of the USDA110 bacteroid proteomes and transcriptomes.** In order to better understand the poor interaction between USDA110 and *A. afraspera*, bacteroid physiology was analyzed through transcriptome and proteome analyses. Efficient soybean bacteroids and free-living USDA110 cells cultivated in rich medium (exponential growth phase under aerobic conditions) were used as references (Fig. 2A).

Prior to quantification of transcript abundances or identification and quantification of protein accumulation, the transcriptome data set was used to reannotate the USDA110 genome with the EugenePP pipeline (23). This allowed the definition of 876 new coding DNA sequences (CDSs), ranging from 92 to 1,091 bp (median size, 215 bp or 71.6 amino acids [aa]), with 11.5% of them having a predicted function or at least a match using InterProScan (IPR). This extends the total number of CDSs in the USDA110 genome to 9,171. Moreover, we also identified 246 noncoding RNAs (ncRNAs), ranging from 49 to 765 bp (median, 76 bp), which were not annotated previously. Proteomic

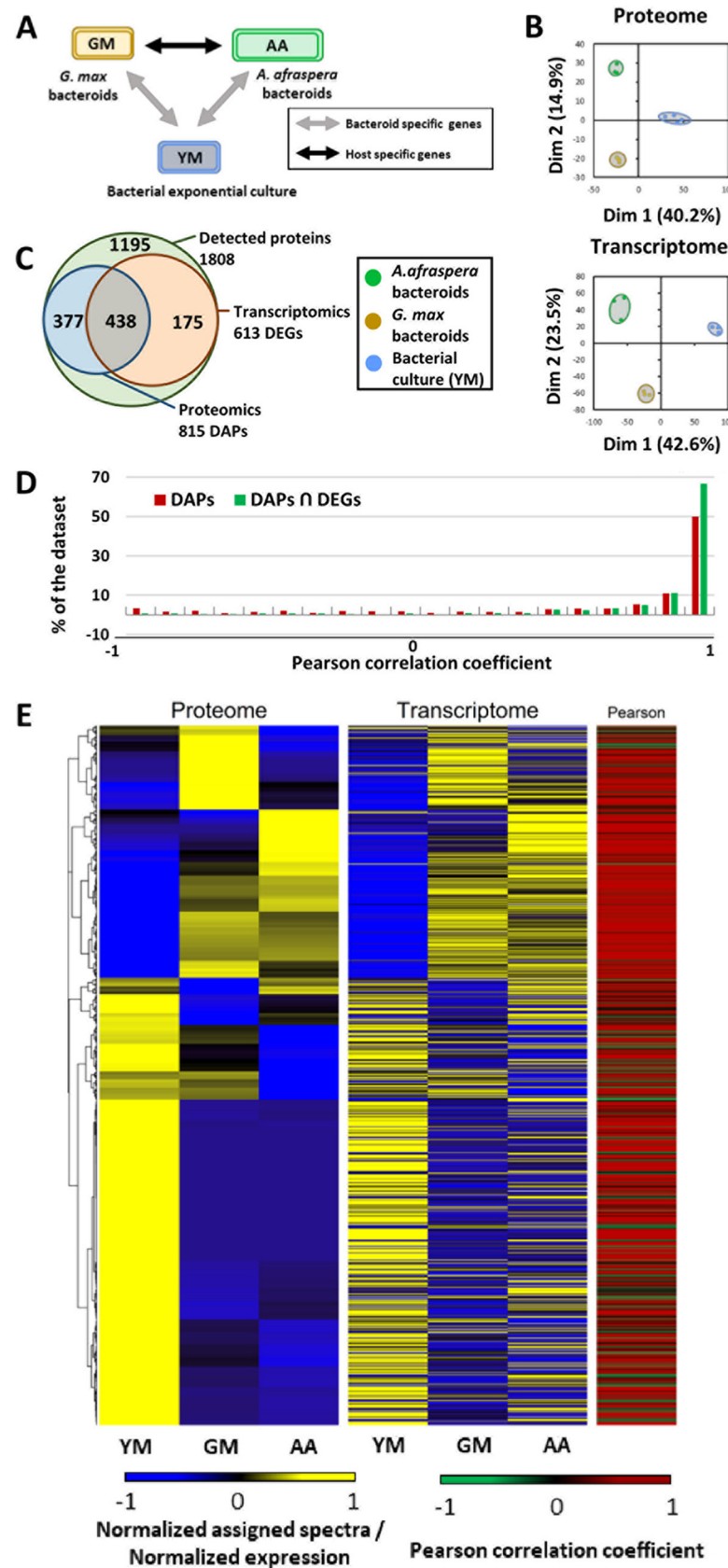

**FIG 2** Experimental setup and general description of the transcriptomics and proteomics data sets. (A) Experimental setup displaying the three biological conditions of this study. (B) Principal-component

evidence could be found for 28 new CDSs (3.2% of the new CDSs; median size, 97.6 aa). The complete reannotation of the genome is described in Table S1.

In the proteome data set, 1,808 USDA110 proteins were identified. Principal-component analysis (PCA) of all the replicate samples and sample types revealed their partitioning according to the tested conditions. The first axis of the PCA (40.2% of the observed variance) separated bacteroid profiles from the exponential-phase culture, whereas the second axis separated *G. max* bacteroids from *A. afraspera* bacteroids (14.9% of the observed variance) (Fig. 2B). The samples of the transcriptome data sets were similarly distributed on the PCA plot, with the first axis explaining 42.6% of the observed variance and the second axis explaining 23.5% of the observed variance (Fig. 2B).

Although differences were less pronounced in the proteome data set than in the transcriptome data set, Clusters of Orthologous Genes (COG) analysis showed similar profiles across functional categories, except for membrane proteins, which were less well identified by proteomics than by transcriptomics (Fig. S4). In the transcriptomic data set, 3,150 genes were differentially expressed under at least one condition (differentially expressed genes [DEGs]). Among the 1,808 proteins identified, 815 showed differential accumulation (differentially accumulated proteins [DAPs]), and 438 of the cognate genes were also differentially expressed in the transcriptome data sets, whereas 175 DEGs were not DAPs (Fig. 2C).

We analyzed the Pearson correlations between transcriptomic and proteomic profiles and found that ~66% of the bacterial functions that showed significant differences by both approaches displayed a high correlation coefficient ($r > 0.9$), whereas <1% of the functions showed a strong negative correlation ($r$ of less than $-0.9$) (Fig. 2D). This observation suggests that the transcriptome (which provides a more exhaustive view than the proteome) and the proteome provide a complementary picture of bacterial physiology, and they tend to show congruent information if we restrict our analysis to the genes with differential accumulation/expression (Fig. 2E). However, there were still ~66% of the DEGs, which were detected by the proteomic analysis, that were not DAPs. Our description of the bacterial functions is primarily based on the functions that were both DEGs and DAPs, as there is stronger evidence of their modulation under the tested conditions. The transcriptome alone is used only when proteomics is not informative, for example, to analyze regulons and stimulons that have been described previously in USDA110.

**Symbiotic functions common to both types of USDA110 bacteroids.** Among the 815 DAPs, 705 and 699 proteins were significantly differentially accumulated in *G. max* and *A. afraspera*, respectively, compared to the bacterial culture control. Strikingly, 646 proteins were commonly differentially accumulated in both plant nodules (Table S1).

In the transcriptomic data set, 1,999 DEGs, representing ~21% of the genome, were identified between the bacterial culture and the bacteroids, regardless of the host. Among them, 1,076 genes displayed higher expression levels in nodules (including 7 newly annotated ncRNAs and 1 newly annotated CDS among the 20 differentially expressed genes with the highest fold changes), and 923 genes were repressed *in planta* (including 2 newly annotated ncRNAs and 2 newly annotated CDSs among the 20 DEGs with the highest fold changes) (Table S1).

Restricting the analysis to the bacterial functions that were both differentially expressed (DEGs) and differentially accumulated (DAPs) *in planta* in both hosts compared to the bacterial culture led to the identification of 222 genes/proteins, with 150

**FIG 2** Legend (Continued)
analysis of the proteomics and transcriptomics data sets. (C) Venn diagram representing the overlap between differentially expressed genes (DEGs) (FDR of <0.01 and |LFC| of >1.58) and differentially accumulated proteins (DAPs) (FDR of <0.05) in at least one comparison and among the population of detected proteins. (D) Pearson correlation coefficient ($r$) distribution between transcriptomic and proteomic data sets based on DAPs only or DAPs that are also DEGs. (E) Heat maps and hierarchical clustering of the 815 DAPs and the corresponding transcriptomic expression values. The heat maps show the standard scores (Z-scores) of assigned spectra and DESeq2-normalized read counts, respectively. The color-coded scale bars for normalized expression and values of Pearson correlation coefficients of the genes are indicated below the heat map. YM, yeast-mannitol culture; GM, *G. max* bacteroids; AA, *A. afraspera* bacteroids.

being upregulated and 72 being repressed *in planta*, respectively (Fig. 3A). Notably, six newly annotated genes were in this gene list, including one putative regulator (Bd110_01119) that was induced during symbiosis. Among the functions common to DEGs and DAPs *in planta*, only four functions showed opposite trends by proteomics and transcriptomics.

The proteome and transcriptome data provided a coherent view of the nitrogen fixation metabolism of *B. diazoefficiens* under the tested conditions. Key enzymes involved in microoxic respiration and nitrogen fixation were detected among the proteins having the highest spectral numbers in the nodule samples (Fig. 3A; Table S1), and the corresponding genes were among the most strongly expressed ones in bacteroids while being almost undetectable under free-living conditions. This includes, for instance, the nitrogenase and the nitrogenase reductase subunits, which constitute the nitrogenase enzyme complex responsible for nitrogen conversion into ammonia. They belong to a locus of 21 genes from *blr1743* (*nifD*) to *bll1778* (putative *ahpC*), including the genes involved in nitrogenase cofactor biosynthesis, electron transport to nitrogenase, and microaerobic respiration, that were among the most highly expressed ones in bacteroids of both host plants, at both the gene and protein expression levels. The slightly higher level of the dinitrogenase reductase NifH detected by proteomics was not supported by Western blot analysis, which showed apparently similar protein levels under both bacteroid conditions (Fig. S5). Strikingly, the two bacteroid types did not show a notable difference in the expression of these genes and proteins, suggesting that the activation of the nitrogen fixation machinery is not a limiting factor underlying the suboptimal efficiency of strain USDA110 in *A. afraspera* nodules.

In addition to these expected bacteroid functions, many other proteins were identified that specifically and strongly accumulated in both nodule types. This is the case for the chaperonins GroS3 and GroL3, which were strongly upregulated and reached high gene expression and protein levels in both bacteroids. The upregulation of these chaperonins is remarkable because other GroEL/GroES proteins (GroS2, GroES, GroL1, GroL6, and GroL7) were also very strongly accumulated in a constitutive manner. This indicates that bacteroids have a high demand for protein folding, possibly requiring specific GroEL isoforms, a situation reminiscent of the requirement of one out of five GroEL isoforms for symbiosis in *Sinorhizobium meliloti*, the symbiont of *Medicago sativa* (12, 24). Another example of a bacteroid-specific function is the hydrogenase uptake system, whose gene expression was induced in both bacteroid types from nearly no expression in culture. The hydrogenase subunit HupL (*bll6941*) was found among the proteins displaying the highest spectral number in the nodule samples, suggesting important electron recycling in bacteroids of the two hosts. Another one is the 1-amino-cyclopropane-1-carboxylic acid (ACC) deaminase (*blr0241*), which was also among the most strongly accumulated proteins in nodules and was significantly less abundant in free-living USDA110. An outer membrane protein (*bll1872*) belonging to the NifA regulon (25) was also strongly induced *in planta*, with a transcript level among the top 10 genes in *A. afraspera*. Additionally, a locus of seven genes (*blr7916* to *blr7922* [*blr7916*–*blr7922*]) encoding an amidase enzyme and a putative peptide transporter composed of two transmembrane domain proteins, two ATPases, and two solute binding proteins was strongly upregulated in the two bacteroid types, with three proteins also being overaccumulated *in planta* (Fig. 3A; Table S1).

Oppositely, motility genes encoding flagellar subunits (*bll5844*–*bll5846*), metabolic enzymes, and transporter subunits were strongly downregulated during symbiosis and hardly detectable at the protein level *in planta* (Fig. 3A).

Taken together, these data show that both bacteroid types displayed a typical nitrogen fixation-oriented metabolism, with a partial shutdown of housekeeping functions. This indicates that despite the apparent reduced symbiotic efficiency of USDA110 in *A. afraspera* nodules, the bacterium fully expressed its symbiotic program within this non-native host as it does in soybean, its original host. Thus, the suboptimal functioning of the *A. afraspera* nodules did not seem to come from a bacterial defect to express the

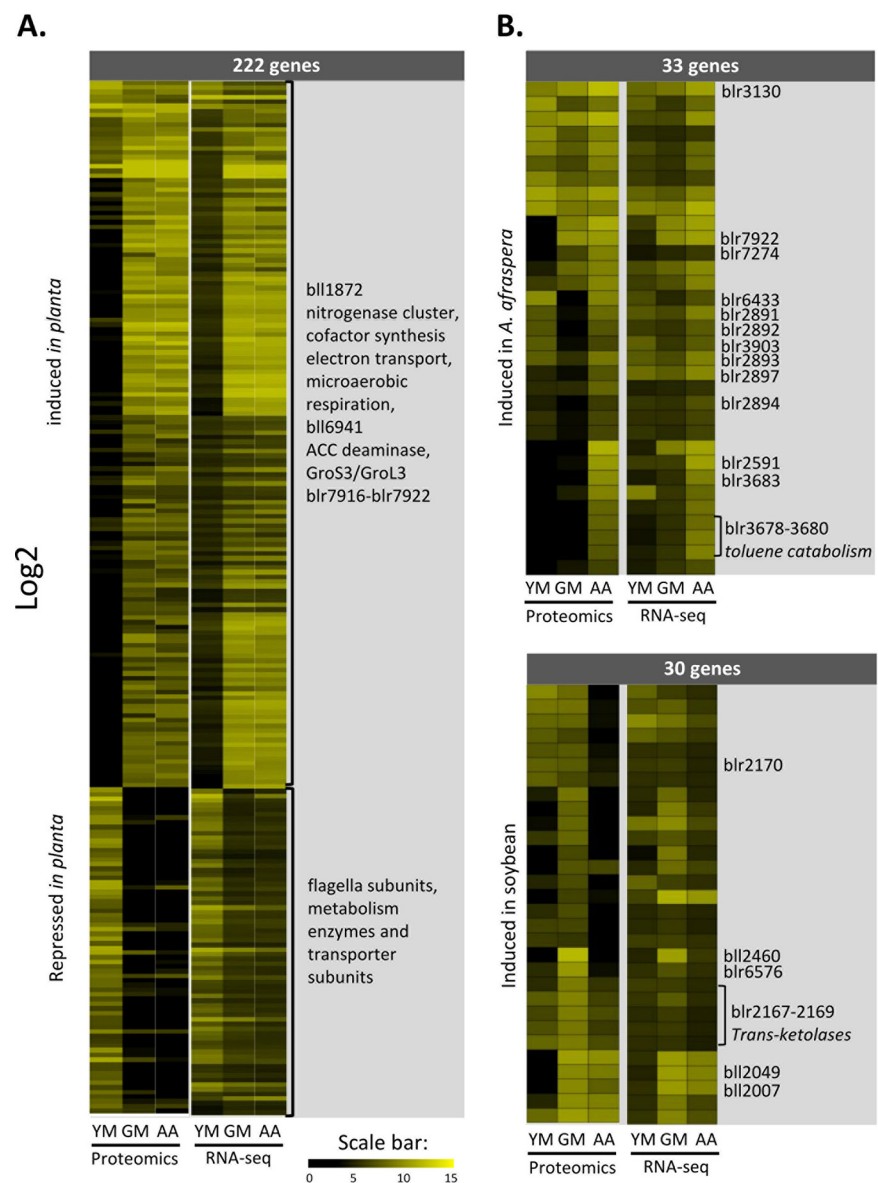

**FIG 3** Symbiosis and host-specific functions that display congruence between transcriptomics and proteomics. (A) Heat map with SOM clustering displaying bacterial functions that are commonly DAPs and DEGs *in planta* in both host plants compared to the culture reference. (B) Heat map displaying bacterial functions that are commonly DEGs and DAPs in one host compared to the other (top, *A. afraspera* > *G. max*; bottom, *G. max* > *A. afraspera*). In panels A and B, data are presented as log₂ fold changes of DESeq2-normalized read counts (RNA-seq) or spectral counts (proteomics). YM, yeast-mannitol culture; GM, *G. max* bacteroids; AA, *A. afraspera* bacteroids.

symbiotic program but possibly came from an unfavorable host microenvironment or a lack of metabolic integration of these maladapted partners.

**Host-specific functions.** Comparison of the *A. afraspera* and *G. max* bacteroids also revealed significant differences in the proteomes and transcriptomes. At the transcriptomic level, 935 DEGs could be identified between the two bacteroid types (509 *A. afraspera* > *G. max* and 426 *G. max* > *A. afraspera*). One notable feature of the transcriptome is the identification of 4 newly annotated ncRNAs and 1 new CDS among the 20 most induced DEGs in *A. afraspera* nodules and the presence of 5 newly annotated CDSs among the 20 most induced DEGs in *G. max* nodules (Table S1). However, when considering only the functions that displayed congruent and significant differences in terms of transcript and protein levels between plant hosts, these fell to 63

genes/proteins, with 33 being induced in *A. afraspera* nodules and 30 being induced in *G. max* nodules (Fig. 3B).

Interestingly, the phenylacetic acid degradation pathway (PaaABCDEIK [*blr2891–blr2897*]) was highly expressed in *A. afraspera* nodules (although only PaaABCD and PaaK have been detected by proteomics), as was an as-yet-uncharacterized cluster of genes putatively involved in toluene degradation (*blr3675–blr3680*). The chaperone GroEL2 was also specifically induced in *A. afraspera*. Similarly, three S1 peptidases (Dop [*blr2591*, *blr3130*, and *blr7274*]) were highly expressed in the nodules of the latter host together with an RND efflux pump (*bll3903*) and an LTXXQ motif protein (*bll6433*), a motif also found in the periplasmic stress response protein CpxP (26). The overaccumulation of these proteins suggests that bacteroids are facing stressful conditions during this interaction with *A. afraspera*. An uncharacterized ABC transporter solute binding protein (*blr7922*) was also overexpressed in *A. afraspera*.

One $\alpha\beta$-hydrolase (*blr6576*) and a TonB-dependent receptor-like protein (*bll2460*) were overaccumulated in a *G. max*-specific manner. Similarly, an uncharacterized metabolic cluster, including transketolases (*blr2167–blr2170*), the heme biosynthetic enzyme HemN1 (*bll2007*), and, to a lesser extent, an anthranilate phosphoribosyltransferase (TrpD, encoded by *bll2049*), was overexpressed in soybean nodules.

**USDA110 transcriptomics data in the perspective of previously described regulons and stimulons.** USDA110 is one of the best-characterized rhizobial strains in terms of transcriptomic responses to various stimuli as well as the definition of regulons (27). We analyzed the behavior of these previously defined gene networks in USDA110 in our data set (Table S2). To initiate the molecular dialog that leads to nodule formation, plants secrete flavonoids like genistein in their root exudates, which are perceived by the rhizobia and trigger Nod factor production. At 14 dpi, when the nodule was formed and functioning, the genistein stimulon, which comprises the NodD1, NodVW, Ttsl, and LafR regulons, was not activated anymore in bacteroids. The symbiotic regulons controlled by NifA, FixK1, FixK2, FixLJ, and sigma 54 (RpoN) were activated *in planta*, indicating that nitrogen fixation was taking place in both hosts. Accordingly, the nitrogen metabolism genes controlled by NtrC were activated *in planta*. Additionally, the PhyR/EcfG regulon involved in the general stress response (GSR) was not activated in bacteroids. However, differences between hosts were not observed for any of these regulons/stimulons. The only stimulon that showed differential expression between hosts is the one involved in aromatic compound degradation, which was highly expressed in *A. afraspera* nodules. A similar upregulation of the vanillate degradation pathway was observed in the transcriptome of *Bradyrhizobium* sp. ORS285 in *A. afraspera* and *Aeschynomene indica* nodules (17), suggesting that dalbergioid hosts display a higher aromatic compound content in nodules than *G. max*. In line with this hypothesis, some of the most differentially accumulated sets of proteins (*A. afraspera* > *G. max*) are involved in the degradation of phenylacetic acid (PaaABCDK and *bll0339*), suggesting that the bacterium converts phenylalanine (or other aromatic compounds) ultimately to fumarate through this route (Fig. 3B) (28). Similarly, enzymes of another pathway involved in phenolic compound degradation (*blr3675–blr3680*) were accumulated in *A. afraspera* nodules (Fig. 3B; Table S1).

**Expression pattern of orthologous genes between ORS285 and USDA110 in *A. afraspera* nodules.** In a previous study (17), a transcriptome analysis was performed on *Bradyrhizobium* sp. ORS285 in interaction with *A. afraspera* and in culture. *Bradyrhizobium* sp. ORS285 is a strain that coevolved with *A. afraspera*, leading to an efficient symbiosis hallmarked by TBD, i.e., cell elongation and polyploidization of the bacteroids. In order to compare the gene expressions of these two nodule-forming rhizobia in culture and *in planta*, we determined the set of orthologous genes between the two strains using the Phyloprofile tool of the MicroScope-MAGE website. This analysis yielded a total of 3,725 genes (Table S3). The heat map in Fig. 4A presents the modulation of gene expression (log$_2$ fold change [LFC]) between *A. afraspera* nodules and the bacterial culture for the orthologous genes in each bacterium, regardless of their statistical significance. When taking a false discovery rate (FDR) of <0.01 into

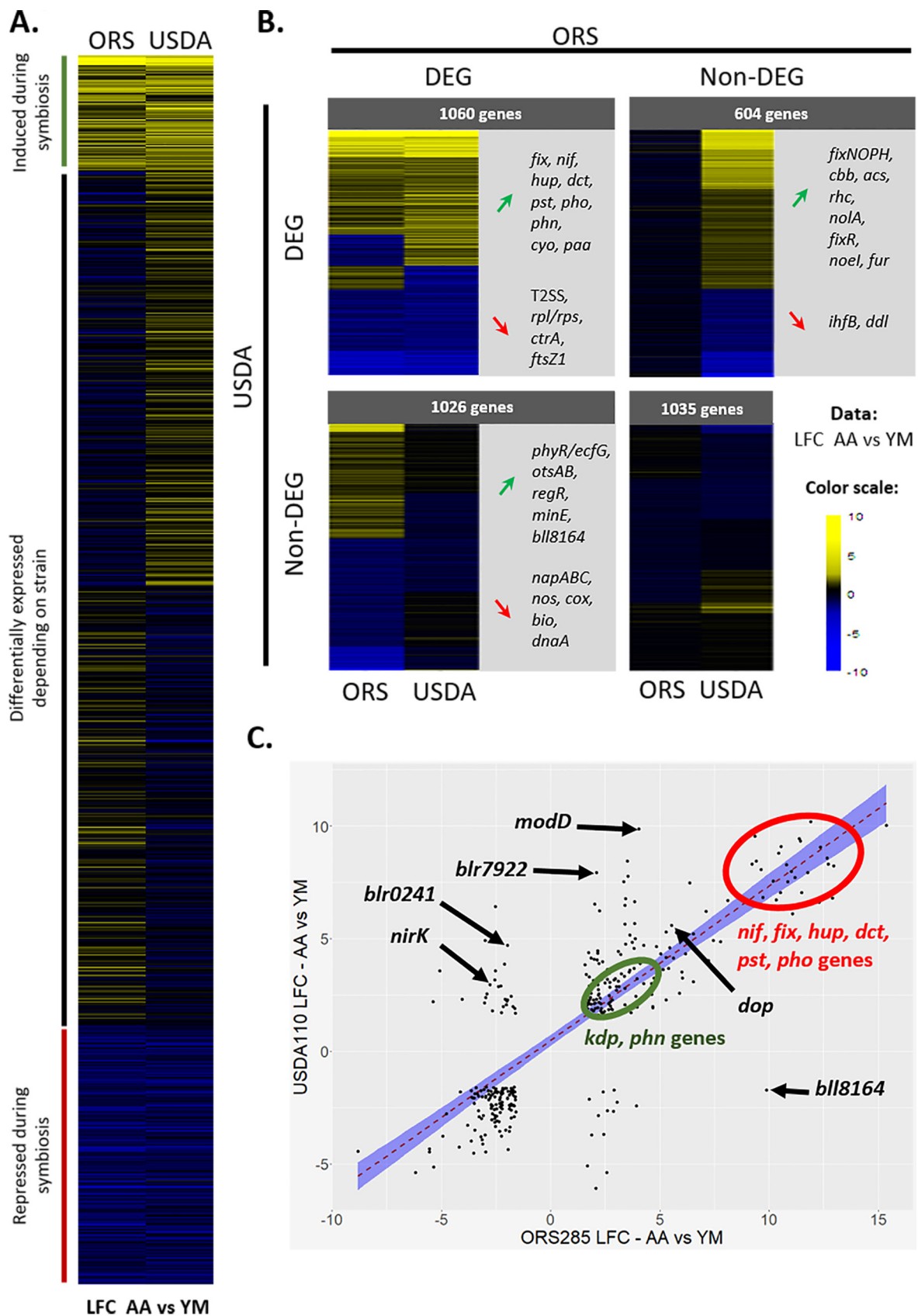

**FIG 4** Expression patterns of *B. diazoefficiens* USDA110 and *Bradyrhizobium* sp. ORS285 orthologous genes *in planta* and in culture. (A) Heat map after SOM clustering of all the orthologous genes of USDA110 and ORS285 obtained with Phyloprofile. Values present the *in planta*

account, we identified sets of genes that were differentially expressed *in planta* in either bacterium or in both bacteria (Fig. 4B).

Only 343 genes displayed differential expression (FDR of <0.01 and |LFC| [absolute LFC] of >1.58) *in planta* in both bacteria compared to their respective culture controls (Fig. 4C). A majority of these genes (86.8%) exhibited congruent expression patterns. First, the *nif*, *fix*, and *hup* genes were commonly and highly induced in both strains during their symbiotic life with *A. afraspera*, a hallmark of a functional symbiosis. However, there were differences in their expression levels, with higher expression levels of the symbiotic genes in ORS285 (*nifHDK* represented 12.5% of all reads in *A. afraspera* nodules) (17) than in USDA110 (*nifHDK* represented only 2.5% of all reads in *A. afraspera* nodules), consistent with a more efficient interaction occurring between ORS285 and *A. afraspera*. Additionally, the Kdp high-affinity transport system, phosphate metabolism (*pstCAB*, *phoU*, *phoE*, and *phoC*), and phosphonate metabolism (*phnHIJKL*) were activated *in planta* in both bacteria (Fig. 4B and C). The stress marker *dop* protease gene was also induced in both bacteria in *A. afraspera* nodules (Fig. 4C).

Additionally, 1,026 genes were differentially expressed solely in ORS285, and similarly, there were 604 DEGs specific to USDA110 (Fig. 4B). For example, the general secretory pathway seemed to be specifically induced in ORS285 (17). Oppositely, USDA110 displayed an induction of the *rhcJQRU* genes, which are involved in the injection of type 3 effector proteins that can be important for the establishment of the symbiosis, whereas they were not induced or even repressed in ORS285 (Fig. 4B). This was also the case for the nitrite reductase-encoding gene *nirK* (*blr7089* [BRAD285_v2_0763]) (Fig. 4C). In addition, USDA110 induced the expression of an ACC deaminase (*blr0241*), while its ortholog was repressed in ORS285 (BRAD285_v2_3570) during symbiosis (Fig. 4C). Bacterial ACC deaminases can degrade ACC, a precursor of ethylene, and thereby modulate ethylene levels in the plant host and promote the nodulation process (29).

***Bradyrhizobium diazoefficiens* USDA110 bacteroids undergo bona fide TBD in *Aeschynomene afraspera* nodules despite very weak morphological and ploidy modifications.** In a previous description of the *A. afraspera-B. diazoefficiens* USDA110 interaction, the typical TBD features were not observed, and the bacteroids were very similar to those in *G. max*, where no TBD occurs (16). At the molecular level, the accumulation of the replication initiation factor DnaA was higher in soybean than in *A. afraspera* (Table S1). Similarly, the MurA peptidoglycan synthesis enzyme (encoded by *bll0822*) that may play a role in cell elongation during TBD was detected at similar levels in both bacteroids (Table S1). Taken together, the molecular data did not clearly indicate whether USDA110 bacteroids undergo TBD in *A. afraspera*. Therefore, we investigated the features of the USDA110 bacteroids in *A. afraspera* nodules in more detail.

We analyzed bacteroid differentiation features in USDA110 bacteroids extracted from soybean and *A. afraspera* nodules. The interaction between *A. afraspera* and *Bradyrhizobium* sp. ORS285 was used as a positive control for TBD features (9, 30, 31). TBD is characterized by cell elongation. We quantified the cell length, width, area, and shape of purified bacteroids and culture controls. Whereas ORS285 bacteroids were enlarged within *A. afraspera* nodules compared to their free-living counterparts, USDA110 bacteroids were similar to those of free-living bacteria in both soybean and *A. afraspera* (Fig. 5A; Fig. S6). Another feature of TBD is endoreduplication. Analysis of the bacterial DNA content of ORS285 bacteroids in *A. afraspera* by flow cytometry shows peaks at 6C and higher (9). As expected, USDA110 bacteroids in *G. max* yielded only two peaks, at 1C and 2C, similarly to the cycling cells in the bacterial culture sample (Fig. 5B) (16). Strikingly, similar results were obtained for USDA110 in *A. afraspera*.

**FIG 4** Legend (Continued)

LFCs calculated for the read counts of the culture control versus *A. afraspera* 14-dpi nodules. (B) Heat maps of the orthologous genes after filtering on the FDR (<0.01) values. Selected genes are highlighted for each class of interest. T2SS, type II secretion system. (C) Dot plot of the orthologous genes that are DEGs (FDR of <0.01 and |LFC| of >1.58) *in planta* (i.e., in *A afraspera* nodules) in both strains. The red dashed line is for the linear regression, and the blue envelope shows the 0.95 confidence interval of the linear regression.

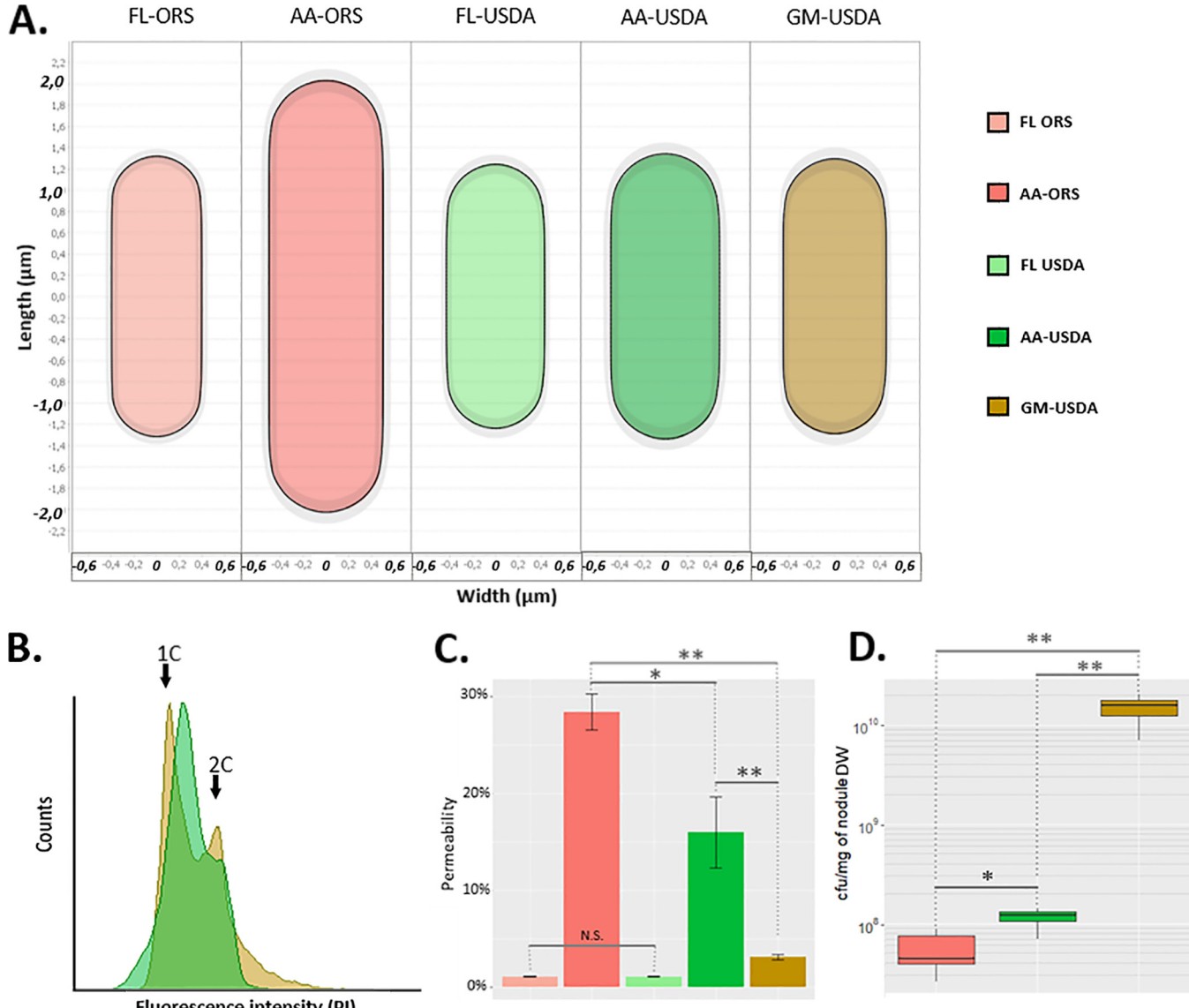

**FIG 5** *B. diazoefficiens* USDA110 displays atypical bacteroid differentiation features in *A. afraspera* nodules. (A) Average cell shape of free-living bacteria and bacteroids determined by MicrobeJ ($900 < n < 21,000$). (B) DNA content of USDA110 bacteroids extracted from soybean and *A. afraspera* determined by flow cytometry. (C) Assessment of the permeability of USDA110 and ORS285 free-living cells and bacteroids 20 min after PI treatment. *, P value of <0.01 by a Wilcoxon test. Five biological replicates were performed for bacteroids, and two were performed for free-living bacteria. (D) Viability of soybean and *A. afraspera* extracted bacteroids at 14 dpi. Asterisks point out significant differences according to a Wilcoxon test. *, P value of <0.05; **, P value of <0.01. Data are representative of results from 10 independent plants. DW, dry weight.

Thus, with respect to the DNA content and cell size, the USDA110 bacteroids did not display the typical TBD features in *A. afraspera* nodules. Loss of membrane integrity is a third hallmark of TBD that likely strongly contributes to the loss of viability of bacteroids. Time course analyses of propidium iodide (PI) uptake by bacteroids and the corresponding culture controls were performed to assess bacteroid permeability (Fig. S7). Twenty minutes after PI application, USDA110 bacteroids from *A. afraspera* displayed an increased permeability that was much closer to that of ORS285 bacteroids in interaction with *A. afraspera* than to the low permeability of USDA110 bacteroids from *G. max* nodules (Fig. 5C). Also, the free-living counterparts exhibited very low permeability. Taken together, these results suggest that the envelope of USDA110 bacteroids is more permeable in the NCR-producing *A. afraspera* nodules, even if it does not reach the permeability level of the ORS285 strain. To analyze bacterial viability, bacteroids extracted from nodules were plated, and the CFU were determined (Fig. 5D).

In *G. max*, USDA110 formed $1.46 \times 10^{10}$ colonies/mg nodule (~100% survival). Oppositely, ORS285 formed only $5.42 \times 10^7$ colonies/mg nodule in *A. afraspera* (~0.5% survival). Interestingly, USDA110 formed $1.13 \times 10^8$ colonies/mg nodule in *A. afraspera* (~1% survival), indicating that despite the absence of cell enlargement and endoreduplication, USDA110 bacteroids lose their viability and undergo bona fide terminal differentiation in *A. afraspera*. Thus, in the NCR-producing plant *A. afraspera*, USDA110 bacteroids displayed a disconnection of the four canonical TBD features (i.e., cell size, ploidy level, membrane permeability, and cell viability).

## DISCUSSION

**Aeschynomene afraspera triggers atypical but terminal differentiation of USDA110 bacteroids.** In a previous study, we noticed that in *A. afraspera*, USDA110 forms a functional symbiosis, although bacteroids do not display features that are usually associated with TBD (16). Here, we show that no endoreduplication and cell elongation of USDA110 occur in terminally differentiated bacteroids that fix nitrogen in a suboptimal way. Accordingly, the protein level of DnaA, the genome replication initiator, was higher in soybean than in *A. afraspera* bacteroids, and the MurA level was not different between bacteroid conditions, confirming that polyploidization and cell elongation did not occur in this host. Such unusual terminal bacteroid differentiation is reminiscent of the bacteroids in *Glycyrrhiza uralensis*. This plant of the IRLC expresses NCR peptides (11). However, one of its compatible symbionts, *Sinorhizobium fredii* strain HH103, does not undergo any loss of viability and shows no change in DNA content and no cell elongation (32), while another symbiont, *Mesorhizobium tianshanense* strain HAMBI 3372, showed all TBD features (33). The influence of the bacterial genotype on the terminal/nonterminal differentiation of bacteroids was also suggested in *Medicago truncatula*, in which the gene *hrrP* might confer to some *Sinorhizobium* strains resistance against the differentiation process triggered by some *M. truncatula* ecotypes (34). In these two IRLC plants (i.e., *M. truncatula* and *G. uralensis*), bacteria undergo complete TBD or no TBD at all in a strain-dependent manner, but there is no clear uncoupling of the features of TBD (cell elongation/endoreduplication/altered viability) as shown here in the case of *B. diazoefficiens* USDA110-*A. afraspera*.

The surprising differentiation of USDA110 in *A. afraspera* nodules raises questions about the molecular mechanisms supporting this phenomenon. We consider two possible hypotheses. First, strain USDA110 might be more sensitive to the differentiation factors of the host than strain ORS285 and be rapidly "terminally" differentiated, before the other differentiation features that are potentially important for symbiotic efficiency can take place. Alternatively, USDA110 might be resistant to the plant effectors that trigger the elongation and polyploidization features.

In agreement with the latter possibility, the application of NCR peptides has a very limited effect on strain USDA110 compared to *S. meliloti* and other plant-associated bacteria (16, 35). NCR insensitivity may be due to the thick hopanoid layer that is present in the outer membrane of strain USDA110, as the hopanoid biosynthesis mutant *hpnH* is more sensitive to NCR peptides and shows symbiotic defects in *A. afraspera* but not in *G. max* (36). Moreover, the altered peptidoglycan structure in the strain USDA110 DD-carboxypeptidase mutant resulted in an increased TBD process with endoreduplicated and elongated bacteroids in *A. afraspera* (16). This suggests that the envelope of strain USDA110 prevents canonical TBD from occurring. Possibly, NCR peptides are not able to reach their intracellular targets required to induce endoreduplication and cell division arrest, while their effect on cell viability through pore formation and membrane destabilization is still effective.

A survey of TBD in the legumes identified multiple occurrences of the process in several subclades of the legumes but found that the majority of legumes do not have TBD (37). The classification in this study was based on a morphological analysis of the bacteroids. Ancestral state reconstruction based on this classification suggested that the nondifferentiated bacteroids are ancestral and that TBD evolved at least five times independently in legumes (37). The discovery of bacteroids that are terminally

differentiated without any obvious morphological changes opens the possibility that the occurrence of TBD might be underestimated in the legume family. Similarly, in the IRLC, the extent of morphological bacteroid differentiation was correlated with the size of the cationic NCR peptide repertoire, and in legumes with few NCR peptides, the morphological modification of bacteroids can be minor (11, 33). In addition, at the molecular level, TBD was originally ascribed to the production of symbiotic antimicrobial peptides, the NCRs, by nodules (7), but more recently, other types of antimicrobial peptides such as the NCR-like, GRP, MBP1, and CAPE peptides specifically produced in nodules of different plants were proposed to contribute to bacteroid differentiation (9, 38–40). Thus, if TBD is indeed more widespread than currently estimated on the basis of morphological bacteroid features, the currently proposed evolutionary scenario of bacteroid formation might require revision.

**Terminal differentiation is associated with specific stress responses.** The TBD of strain USDA110 in *A. afraspera* is associated with a high accumulation of stress markers compared to the *G. max* bacteroids. These markers include four proteases (Dop, Lon-*blr6174*, *blr3130*, and *blr7274*) and one chaperonin (GroL4). Similar inductions of proteases and chaperonins have been reported in NCR-treated *S. meliloti* cultures (35), indicating that this response may be linked to the perception of *A. afraspera* NCR-like peptides in USDA110.

The genes encoding these stress-related proteins are not part of the well-characterized general stress response (GSR) controlled by the PhyR/EcfG signaling cascade in *B. diazoefficiens* USDA110 (41). On the other hand, we found that the PhyR/EcfG regulon in USDA110 is not activated in the bacteroids of both host plants (see Table S2 in the supplemental material). This observation contrasts with our previous study of the *Bradyrhizobium* sp. ORS285 transcriptome during symbiosis with *Aeschynomene* plants, which showed that the PhyR/EcfG cascade was upregulated *in planta* (17). Nevertheless, the expression of the Dop protease was induced in *A. afraspera* in both bacteria (Fig. 4C). Together, the omics data suggest that bacteroids of *Bradyrhizobium* spp. activate stress-related genes in the TBD-inducing *A. afraspera* host but that differences exist in the activation of specific stress responses at the strain level.

**Correlation between bacteroid differentiation features and symbiotic efficiency for the plant.** TBD is associated with the massive production of symbiotic antimicrobial peptides such as NCR, NCR-like, and CAPE peptides in different plants (5, 9, 38, 40). They represent ~10% of the nodule transcriptomes in *M. truncatula* (analysis of the data from reference 42), and their production is thus potentially a strong energetic cost for the plant, raising questions about the benefits of the TBD process. TBD appeared independently in different legume clades (9, 37), suggesting that plants imposing this process obtain an advantage that might be a higher symbiotic benefit. Increased symbiotic efficiency has indeed been observed in hosts imposing TBD (17, 43, 44). The findings reported here, comparing bacteroids and symbiotic efficiencies in *A. afraspera* infected with strain ORS285 and strain USDA110, are in agreement with this hypothesis. Also, in the symbiosis of *M. truncatula* in interaction with different *S. meliloti* strains, a similar correlation was observed between the level of bacteroid differentiation and plant growth stimulation (45). However, the simultaneous analysis of the bacteroid differentiation and symbiotic performances of an extended set of *Aeschynomene-Bradyrhizobium* interactions has shown that, perhaps not unexpectedly, the symbiotic efficiency of the plant-bacterium couple is not correlated solely with bacteroid differentiation and that other factors can interfere with the symbiotic efficiency as well (46).

**Conclusion.** *Bradyrhizobium diazoefficiens* USDA110 is a major model of legume-rhizobium symbiosis, mainly thanks to its interaction with *G. max*, the most cultivated legume worldwide. Although omic studies have been conducted in this strain in symbiosis with various hosts (13, 25), this is the first time that this bacterium has been studied at the molecular level in symbiosis with an NCR-producing plant that normally triggers typical terminal bacteroid differentiation in its symbionts. The symbiosis between USDA110 and *A. afraspera* is functional even if nitrogen fixation and plant benefits are suboptimal.

Terminal bacteroid differentiation is taking place in the NCR-producing host *A. afraspera*, as bacterial viability is impaired in USDA110 bacteroids, whereas morphological changes and the cell cycle switch to endoreduplication are not observed. We also show by combining proteomics and transcriptomics that the bacterial symbiotic program is expressed in *A. afraspera* nodules in a way similar to that in *G. max*, although host-specific patterns were also identified. However, the bacterium is under stressful conditions in the *A. afraspera* host, possibly due to the production of NCR-like peptides in this plant. The integration of data sets from different bacteria in symbiosis with a single host, like ORS285 and USDA110 in symbiosis with *A. afraspera*, shed light on the differences in the stress responses activated in *A. afraspera* and confirmed that the symbiosis is functional but suboptimal in this interaction. The molecular data presented here provide a set of candidate functions that could be analyzed for their involvement in adaptation to a new host and to the TBD process.

## MATERIALS AND METHODS

**Bacterial cultures and bacteroid extraction.** *B. diazoefficiens* USDA110 (47) and *Bradyrhizobium* sp. ORS285 were cultivated in yeast-mannitol (YM) culture medium at 30°C in a rotary shaker (48). For transcriptomic analysis, culture samples (optical density at 600 nm [$OD_{600}$] = 0.5) were collected and treated as described previously by Chapelle et al. (49).

*G. max* ecotype Williams 82 and *A. afraspera* seeds were surface sterilized, and the plants were cultivated and infected with rhizobia for nodule formation as described previously by Barrière et al. (16). Nodules were collected at 14 days postinoculation (dpi), immediately immersed in liquid nitrogen, and stored at −80°C until use. Each tested condition (in culture and *in planta*) was produced in biological triplicates.

**Phylogeny analysis.** Nucleotide sequences of *matK* genes were collected from the NCBI database using accession numbers described previously (50, 51) and analyzed on by Phylogeny.fr (www .phylogeny.fr). They were aligned using ClustalW with manual corrections, before running a phyML (GTR-gamma model) analysis with 500 bootstraps. A Bayesian inference tree was also generated (GTR+G+I) and provided a topology similar to the one for the maximum likelihood (ML) tree (data not shown). Trees were visualized and customized using TreeDyn.

**Genome annotation and RNA-seq analysis.** Nodule and bacterial culture total RNA was extracted and treated as previously described (17). Oriented (strand-specific) libraries were produced using the SOLiD total RNA-seq kit (Life Technologies) and sequenced on a SOLiD 3 station, yielding ∼40 million 50-bp single reads. Trimming and normalization of the reads were performed using CLC workbench software. Subsequently, the reads were used to annotate the genome using EuGenePP (23), and mapping was performed using this new genome annotation. Analysis of the transcriptome using DESeq2 and data representation were performed as previously described (17). Differentially expressed genes (DEGs) showed an absolute $\log_2$ fold change (|LFC|) of >1.58 (i.e., fold change of >3) with a false discovery rate (FDR) of <0.01.

**Proteomic analysis.** Bacteroids were extracted from 14-dpi frozen nodules (6), while bacterial culture samples were collected as described above, and the bacterial pellets were resuspended in −20°C acetone and lysed by sonication. Protein solubilization, dosage, digestion (2% [wt/wt] trypsin), and solid-phase extraction (using a Phenomenex polymeric $C_{18}$ column) were performed as described previously (52). Peptides from 800 ng of proteins were analyzed by LC-MS/MS with a Q Exactive mass spectrometer (Thermo Electron) coupled to a nanoLC Ultra 2D instrument (Eksigent) using a nanoelectrospray interface (noncoated capillary probe, 10-$\mu$m internal diameter [ID]; New Objective). Peptides were loaded on a Biosphere $C_{18}$ trap column (particle size of 5 $\mu$m, pore size of 12 nm, inner/outer diameters of 360/100 $\mu$m, and length of 20 mm; NanoSeparations) and rinsed for 3 min at 7.5 $\mu$l/min of 2% acetonitrile (ACN)–0.1% formic acid (FA) in water. Peptides were then separated on a Biosphere $C_{18}$ column (particle size of 3 $\mu$m, pore size of 12 nm, inner/outer diameters of 360/75 $\mu$m, and length of 300 mm; NanoSeparations) with a linear gradient from 5% buffer B (0.1% FA in ACN) and 95% buffer A (0.1% FA in water) to 35% buffer B and 65% buffer A in 80 min at 300 nl/min, followed by a rinsing step at 95% buffer B and 5% buffer A for 6 min and a regeneration step with parameters from the start of the gradient for 8 min. Peptide ions were analyzed using Xcalibur 2.1 software in the data-dependent mode with the following parameters: (i) a full MS scan was acquired for the *m/z* 400 to 1,400 range at a resolution of 70,000 with an automatic gain control (AGC) target of 3 × $10^6$, and (ii) an $MS^2$ scan was acquired at a resolution of 17,500 with an AGC target of 5 × $10^4$, a maximum injection time of 120 ms, and an isolation window of 3 *m/z*. The normalized collision energy was set to 27. The $MS^2$ scan was performed for the eight most intense ions in the previous full MS scan with an intensity threshold of 1 × $10^3$ and a charge of between 2 and 4. Dynamic exclusion was set to 50 s. After conversion to mzXML format using msconvert (3.0.3706) (53), data were searched using X!Tandem (version 2015.04.01.1) (54) against the USDA110 reannotated protein database and a homemade database containing current contaminants. In a first pass, trypsin was set to strict mode; cysteine carbamidomethylation was set as a fixed modification; and methionine oxidation, protein N-terminal acetylation with or without protein N-terminal methionine excision, N-terminal glutamine and carbamidomethylated cysteine deamidation, and N-terminal glutamic dehydration were set as potential modifications. In a refine pass, semienzymatic peptides were allowed. Protein inference was performed using X!TandemPipeline (version 3.4.3) (55). A protein was

validated with an E value of $<10^{-5}$ and 2 different peptides with an E value of $<0.05$. Proteins from the contaminant database (*Glycine max* proteins and unpublished *Aeschynomene* expressed sequence tags) were removed after inference. Proteins were quantified using the spectral counting method (56). To discriminate differentially accumulated proteins (DAPs), analysis of variance (ANOVA) was performed on the spectral counts, and proteins were considered a DAP when the $P$ value was $<0.05$.

**Metabolomic analysis.** Metabolites and cofactors were extracted from lyophilized nodules and analyzed by gas chromatography-mass spectrometry (GC-MS) and LC-MS, respectively, according to methods described previously by Su et al. (57) and Guérard et al. (58).

**Plant biomass and nitrogen fixation analysis.** Dry masses of the shoot, root, and nodules were measured, and the shoot/root mass ratio was calculated. The mass gain per gram of dry nodule was calculated as the difference between the total mean masses of the plants of interest and those of the noninoculated plants, divided by the mean mass of nodules. Thirty plants were used under each condition. Nitrogenase activity was assessed by an acetylene reduction assay (ARA) on 10 plants under each condition as previously described (31). The elemental analysis of leaf carbon and nitrogen contents was performed as described previously (18).

**Analysis of *B. diazoefficiens* USDA110 regulons and stimulons.** Gene sets defined as regulons and stimulons were collected from the literature, and the regulons/stimulons were considered activated/repressed when $\geq$40% of the corresponding genes were DEGs in a host plant compared to the culture conditions.

**Comparison of orthologous gene expression between *B. diazoefficiens* USDA110 and *Bradyrhizobium* sp. ORS285.** The list of orthologous genes between USDA110 and ORS285 was determined using the Phyloprofile tool of the MicroScope-MAGE platform (59), with an identity threshold of 60%, a maxLrap of $>0$, and a minLrap of $>0.8$. The RNA-seq data from a previous study (17) and those of this study were used to produce heat maps for the genes displaying an FDR of $<0.01$ (*A. afraspera* versus YM medium) using R (v3.6.3) and drawn using pheatmap (v1.0.12) coupled with kohonen (v3.0.10) for gene clustering using the self-organizing maps (SOM) method. The DEGs in both organisms (*A. afraspera* versus YM medium) were plotted for USDA110 and ORS285.

**Analysis of TBD features.** Bacteroids were extracted from 14-dpi nodules and analyzed using a CytoFLEX S instrument (Beckman-Coulter) (31). For ploidy and live/dead analyses, samples were stained with propidium iodide (PI) (Thermo Fisher) (50-$\mu$g $\cdot$ ml$^{-1}$ final concentration) and Syto9 (Thermo Fisher) (1.67 $\mu$M final concentration). PI permeability was assessed over time on live bacteria. *Bradyrhizobium* sp. ORS285.pMG103-*nptII-GFP* (30) and *B. diazoefficiens* USDA110 sYFP2-1 (60) strains were used to distinguish bacteroids from debris during flow cytometry analysis. For each time point, the suspension was diluted 50 times for measurement in the flow cytometer. The percentage of bacteroids permeable to PI was estimated as the ratio of PI-positive over total bacteroids (green fluorescent protein [GFP]/yellow fluorescent protein [YFP] positive). Heat-killed bacteroids were used as a positive control to identify the PI-stained bacteroid population.

For bacteroid viability assays, nodules were collected and surface sterilized (1 min of 0.4% NaClO, 1 min of 70% ethanol, and two washes in sterile water). Bacteroids were subsequently prepared as previously described (31) and serially diluted and plated (5 $\mu$l per spot) in triplicate on YM medium containing 50 $\mu$g $\cdot$ ml$^{-1}$ carbenicillin. CFU were counted 5 days after plating and divided by the total nodule mass.

Bacterial cell shape, length, and width were determined using confocal microscopy image analysis. Bacteroid extracts and stationary-phase bacterial cultures were stained with 2.5 nM Syto9 for 10 min at 37°C and mounted between a slide and a coverslip. Bacterial imaging was performed on an SP8 confocal laser scanning microscope (Leica Microsystems) equipped with hybrid detectors and a 63$\times$ oil immersion objective (Plan Apo, 1.4 numerical aperture [NA]; Leica). Under each condition, multiple z-stacks (2.7-$\mu$m width and 0.7-$\mu$m step) were automatically acquired (excitation at 488 nm and collection of fluorescence at 520 to 580 nm).

Prior to image processing, each stack was transformed as a maximum-intensity projection using ImageJ software (https://imagej.nih.gov/ij/). Bacterial detection was performed with MicrobeJ (https://www.microbej.com/) (61). First, bacteria were automatically detected on every image using an intensity-based thresholding method with a combination of morphological filters (area, 1 to 20 $\mu$m$^2$; length, 1 $\mu$m to $\infty$; width, 0.5 to 1.3 $\mu$m), and every object was fitted with a "rod-shaped" bacterial model. To ensure high data quality, every image was manually checked to remove false-positive results (mainly plant residues) and include rejected objects (mainly fused bacteria). Next, the morphology measurements and figures were directly extracted from MicrobeJ. ORS285 cells in culture and in symbiosis with *A. afraspera* were used as references for the analysis of TBD features.

**Western blot analysis.** Detection of NifH by Western blotting was performed using a commercial polyclonal antibody against a NifH peptide (Agrisera). Western blotting was carried out as previously described (62), using bacterial exponential (OD$_{600}$ = 0.5)- and stationary (OD$_{600}$ $>$ 2.5)-phase cultures as well as 14-dpi nodule-extracted bacteroids.

**Data availability.** Genome annotation of *Bradyrhizobium diazoefficiens* USDA110 using EuGenePP is available at http://doi.org/10.25794/reference/d56qaddg. Transcriptome data have been deposited in the NCBI Gene Expression Omnibus (GEO) and are accessible through GEO series accession number GSE163004. Proteome data are available at http://moulon.inra.fr/protic/usda110_bacteroid_differentiation (login/passwords will be provided upon request).

## SUPPLEMENTAL MATERIAL

Supplemental material is available online only.

**FIG S1**, TIF file, 0.4 MB.

**FIG S2**, TIF file, 0.3 MB.

**FIG S3**, TIF file, 1.2 MB.
**FIG S4**, TIF file, 0.7 MB.
**FIG S5**, TIF file, 0.4 MB.
**FIG S6**, TIF file, 0.7 MB.
**FIG S7**, TIF file, 0.4 MB.
**TABLE S1**, XLSX file, 2.5 MB.
**TABLE S2**, XLSX file, 0.01 MB.
**TABLE S3**, XLSX file, 0.7 MB.

## ACKNOWLEDGMENTS

We thank Dora Latinovics for the production and sequencing of the RNA-seq libraries and Mélisande Blein-Nicolas for her advice regarding the statistical analysis of the proteomic data set.

Quentin Nicoud and Florian Lamouche were supported by a Ph.D. fellowship from the Université Paris-Sud. The present work has benefited from the core facilities of Imagerie-Gif (http://www.i2bc.paris-saclay.fr), a member of IBiSA (http://www.ibisa.net), supported by France-BioImaging (ANR-10-INBS-04–01), and the Labex Saclay Plant Sciences (ANR-11-IDEX-0003-02). This work was funded by the Agence Nationale de la Recherche, grant no. ANR-13-BSV7-0013 and ANR-17-CE20-0011, and used resources from the National Office for Research, Development and Innovation of Hungary, grant no. 120120 to Attila Kereszt.

Quentin Nicoud, Florian Lamouche, Peter Mergaert, Benjamin Gourion, and Benoit Alunni designed the work. Quentin Nicoud, Florian Lamouche, Anaïs Chaumeret, Thierry Balliau, Mickaël Bourge, Florence Guérard, Erika Sallet, Solenn Tuffigo, and Olivier Pierre performed the experiments. Quentin Nicoud, Florian Lamouche, Mickaël Bourge, Erika Sallet, Yves Dessaux, Bertrand Gakière, Françoise Gilard, Istvan Nagy, Attila Kereszt, Michel Zivy, Peter Mergaert, Benjamin Gourion, and Benoit Alunni analyzed the data. Quentin Nicoud, Florian Lamouche, Peter Mergaert, Benjamin Gourion, and Benoit Alunni wrote the paper.

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
