## [Reviewer comments · mSystems]

***Bradyrhizobium diazoefficiens* USDA110 nodulation of *Aeschynomene afraspera* is associated with atypical terminal bacteroid differentiation and suboptimal symbiotic efficiency**

Quentin Nicoud, Florian Lamouche, Anaïs Chaumeret, Thierry Balliau, Romain Le Bars, Mickaël Bourge, Fabienne Pierre, Florence Guérard, Erika Sallet, Solenn Tuffigo, Olivier Pierre, Yves DESSAUX, Françoise Gilard, Bertrand Gakière, Istvan Nagy, Attila Kereszt, Michel Zivy, Peter Mergaert, Benjamin Gourion, and Benoit Alunni

Corresponding Author(s): Benoit Alunni, Université Paris-Saclay

Review Timeline:

Submission Date:	November 24, 2020
Editorial Decision:	January 5, 2021
Revision Received:	March 3, 2021
Accepted:	April 14, 2021

Editor: Michelle Heck

Reviewer(s): Disclosure of reviewer identity is with reference to reviewer comments included in decision letter(s). The following individuals involved in review of your submission have agreed to reveal their identity: Joel S Griffiths (Reviewer #2)

Transaction Report:

DOI: <https://doi.org/10.1128/mSystems.01237-20>

January 5, 2021

Dr. Benoit Alunni
Université Paris-Saclay
I2BC - CEA-CNRS-Université Paris Sud
Bat 34. Avenue de la Terrasse
Gif-sur-Yvette 91198
France

Re: mSystems01237-20 (*Bradyrhizobium diazoefficiens* USDA110 nodulation of *Aeschynomene afraspera* is associated with atypical terminal bacteroid differentiation and suboptimal symbiotic efficiency)

Dear Dr. Benoit Alunni:

Thank you for submitting your paper to mSystems. I really enjoyed this paper and am delighted to accept it with minor revisions. Regarding the comments of reviewer 1 and the need to clarify the display of the proteome and transcriptome data, I do not agree and suggest you leave the figures as they appear in this draft. It is OK to disagree with the reviewer and provide an explanation as to why you disagree. I think the global phenotypic differences at the omics level, which are a valuable part of the story as pointed out by reviewer 2, are evident from these figures. The experiment showing normalized acetylene reduction data is a good idea if it is feasible to do with the time and resources available, but this experiment will not be required for the resubmission to be accepted.

Below you will find the comments of the reviewers.

To submit your modified manuscript, log onto the eJP submission site at <https://msystems.msubmit.net/cgi-bin/main.plex>. If you cannot remember your password, click the "Can't remember your password?" link and follow the instructions on the screen. Go to Author Tasks and click the appropriate manuscript title to begin the resubmission process. The information that you entered when you first submitted the paper will be displayed. Please update the information as necessary. Provide (1) point-by-point responses to the issues raised by the reviewers as file type "Response to Reviewers," not in your cover letter, and (2) a PDF file that indicates the changes from the original submission (by highlighting or underlining the changes) as file type "Marked Up Manuscript - For Review Only."

Due to the SARS-CoV-2 pandemic, our typical 60 day deadline for revisions will not be applied. I hope that you will be able to submit a revised manuscript soon, but want to reassure you that the journal will be flexible in terms of timing, particularly if experimental revisions are needed. When you are ready to resubmit, please know that our staff and Editors are working remotely and handling submissions without delay. If you do not wish to modify the manuscript and prefer to submit it to another journal, please notify me of your decision immediately so that the manuscript may be formally withdrawn from consideration by mSystems.

If your manuscript is accepted for publication, you will be contacted separately about payment when the proofs are issued; please follow the instructions in that e-mail. Arrangements for payment must be made before your article is published. For a complete list of **Publication Fees**, including

supplemental material costs, please visit our website.

Sincerely,

Michelle Heck

Editor, mSystems

Journals Department
Reviewer comments:

Reviewer #1 (Comments for the Author):

Nicoud et al. present a study describing the interaction between the soybean symbiont *Bradyrhizobium diazoefficiens* USDA 110 and *Aeschynomene afraspera*, an alternative host which with *B. diazoefficiens* had little to no coevolution. The authors present three main points in their work: i) Despite forming a functioning symbiosis with nitrogen-fixing bacteroids in root nodules, the association is less efficient compared to a native coevolved photosynthetic bradyrhizobium (strain ORS285); ii) *Aeschynomene* spp. encode NCR peptides and enforce terminal bacteroid differentiation in their microsymbionts. However, USDA 110 only displays selected features of this process compared to the coevolved strain ORS285; iii) Using a combination of metabolomics, proteomics, and transcriptomics the authors describe global changes such as stress responses of USDA 110 when interacting with either its native host soybean or *A. afraspera*. Moreover, the authors also compare USDA 110 and ORS285 bacteroids in *A. afraspera*. This is overall a strong piece of work, however, whilst I find the data presented for points ii and iii very convincing, the manuscript will benefit from strengthening point i. Furthermore, the presentation of the omics datasets could be improved.

Major points

L116 and Fig 1D, E: The authors are making the point that *B. diazoefficiens* USDA 110 is poorly adapted to the *A. afraspera* nodule environment and bacteroids are less efficient (compared to ORS285) in nitrogen fixation (as the measurements of nitrogen export products suggest). From this aspect, I find the selection of normalization used in panels D and E counterintuitive. Acetylene reduction is a direct proxy for nitrogenase activity and normalization to nodule fresh or dry weight usually results in a good estimate of bacteroid efficiency. On the other hand, mass gain (shoot or whole plant dry weight) is a good measurement for the overall efficiency of the symbiosis. Thus, I

highly suggest swapping the normalization (per nodule mass for acetylene reduction and biomass per plant) or at least also indicating absolute nodule mass in addition to the existing acetylene reduction assay graph.

I have no doubts that ORS285 is overall the better symbiont for *A. afraspera* than USDA 110 (based on the data presented in Figs 1F, S1, and S2). However, from the currently presented data it is not entirely clear to me, whether USDA 110 is performing poorly in *A. afraspera* or whether ORS285 is the generally more efficient symbiont. The performance of USDA 110 seems very comparable between soybean and *A. afraspera*. It must also be noted here that nodulation kinetics are different for different legumes. 14 dpi for soybean seems a rather early time point and depending on the growth conditions most reports use at least 3-4 weeks old plants. Ideally, the authors could present normalized acetylene reduction data for several time points depending on the nodulation kinetics of the different hosts (e.g. 14 and 21 dpi for soybean and developmentally comparable time points for *A. afraspera*).

L225-232: It was previously shown that groESL genes in *B. diazoefficiens* are differentially regulated. Whilst groESL2 (weak) and 4 (strong) are constitutively expressed, groESL1 is upregulated upon heat shock and groESL3 expression is NifA dependent, thus showing higher expression under symbiotic and micro/anaerobic conditions (Fischer HM et al. 1993 EMBO J 127:2901-2912). Do the authors use the same gene nomenclature and if so, they should discuss this found discrepancy of GroESL isoform regulations with the previous study.

Figs 3, 4, and S4: I find the current data presentation of the proteomics and transcriptomics data in heat maps unintuitive and hard to interpret. In its current form, the reader is presented with large heatmaps of. Of the nature of the underlying proteins and genes only rough associations can be made (e.g. upregulated in plants). Only occasionally, the identity of the underlying gene is revealed (Fig 3B) whilst in most cases information is lacking or the indicated examples are not clearly assigned to a row of the heat map. As such, in most cases I do not know what to make of the individual rows. I suggest the authors find alternative or complementary visualization methods similar to Fig 4C (which should be mentioned in the Figure legend), e.g., Venn diagrams for overall amounts of DAPs and DEGs or Volcano plots if the quantitative differences should be retained. Since such diagrams are likewise only giving a general idea of the overall datasets, additional information of the DEGs and DAPs of each category according to their function might be helpful, e.g. incorporating Panels similar to Fig S4 for each category.

Minor points

The authors occasionally switch from past to present tense when describing the results, please maintain past tense throughout the manuscript.

L42743: '... involving ... membrane permeability ...' reads strange, I suggest changing to 'increased membrane permeability' or 'membrane permeability changes' or so.

L46: establishes

L57: into the biosphere

L62: Please reconsider the wording of non-natural, which implies a synthetic or artificial interaction. However, the study deals with two wild-type organisms entering a functional (although poorly efficient) symbiosis.

L118: Space between 'seemed' and 'also'

L119 and Fig S1: How many plants (replicates) were used for the elemental analysis and why are there no error bars on the nitrogen content columns? Did the replicates result in such consistent results that the error bars cannot be drawn?

L137: Please provide a reference for the observation of phosphate accumulation in inefficient nodules.

L272 and Fig 4: Online 272, the authors refer to the blr3675-80 cluster as encoding toluene degradation enzymes, but in Fig 4, the cluster (blr3678-80) is annotated as toluene anabolism.

L275: There seems a word missing about CpxP.

Fig S3: What is the meaning of metabolite xy 1 and xy 2 (e.g. glucose 1 and glucose 2; glutamine 1 and glutamine 2)? Are these different derivatives from GC-MS sample preparation? If so, could they be pooled?

Reviewer #2 (Comments for the Author):

This manuscript reports a comprehensive description of the symbiosis between the soybean-adapted Bradyrhizobium strain USDA110 and the less compatible legume host *Aeschynomene afraspera*. *Aeschynomene* is a host that induces terminal bacteroid differentiation, and this is morphologically obvious with its adapted bradyrhizobial microsymbiont strain ORS285. However, with USDA110, *Aeschynomene* induces terminal differentiation without such obvious morphological effects, and nitrogen fixation in this case is somewhat attenuated. This paper explores this maladapted interaction, and seeks to understand distinctions between more and less compatible symbioses based on proteomic, transcriptomic, and metabolomic data. A key comparison was USDA110 response to the more compatible soybean host and the less compatible *Aeschynomene* host.

An important by-product of this work was the re-annotation of USDA110, including 876 new protein-coding genes and 246 non-coding RNAs.

In general, the authors report reasonably good agreement between transcriptomic and proteomic outcomes in terms of genes differentially expressed (or proteins differentially accumulated) in nodules versus free-living conditions. Many USDA110 genes are induced in both plant hosts, with many of these genes being poorly characterized in terms of biochemical functions.

Most interesting is the set of nearly one thousand USDA110 genes that are preferentially induced on one plant host or the other. Of these several dozen differences are corroborated by the proteomic data. For example, USDA110 appears to deploy aromatic compound degradation and stress-response functions specifically in the maladapted host (*Aeschynomene*).

The authors are careful to test whether USDA110 is truly terminally differentiated within *Aeschynomene* nodules, showing that while the vast majority of bacteroids are not recoverable in free-living conditions, they do not exhibit other traits characteristic of terminally differentiated bacteroids (such as increased size and endoreduplicated DNA).

The paper provides a valuable compendium of additional data for an already carefully studied symbiotic bacterial strain (USDA110), and provides some insight into how its interaction with two distantly related legume hosts differs at the level of gene expression and biochemical signatures.

The work appears to be well carried out with appropriate computational and statistical analyses.

Nicoud et al. present a study describing the interaction between the soybean symbiont *Bradyrhizobium diazoefficiens* USDA 110 and *Aeschynomene afraspera*, an alternative host which with *B. diazoefficiens* had little to no coevolution. The authors present three main points in their work: i) Despite forming a functioning symbiosis with nitrogen-fixing bacteroids in root nodules, the association is less efficient compared to a native coevolved photosynthetic bradyrhizobium (strain ORS285); ii) *Aeschynomene* spp. encode NCR peptides and enforce terminal bacteroid differentiation in their microsymbionts. However, USDA 110 only displays selected features of this process compared to the coevolved strain ORS285; iii) Using a combination of metabolomics, proteomics, and transcriptomics the authors describe global changes such as stress responses of USDA 110 when interacting with either its native host soybean or *A. afraspera*. Moreover, the authors also compare USDA 110 and ORS285 bacteroids in *A. afraspera*. This is overall a strong piece of work, however, whilst I find the data presented for points ii and iii very convincing, the manuscript will benefit from strengthening point i. Furthermore, the presentation of the omics datasets could be improved.

Major points

L116 and Fig 1D, E: The authors are making the point that *B. diazoefficiens* USDA 110 is poorly adapted to the *A. afraspera* nodule environment and bacteroids are less efficient (compared to ORS285) in nitrogen fixation (as the measurements of nitrogen export products suggest). From this aspect, I find the selection of normalization used in panels D and E counterintuitive. Acetylene reduction is a direct proxy for nitrogenase activity and normalization to nodule fresh or dry weight usually results in a good estimate of bacteroid efficiency. On the other hand, mass gain (shoot or whole plant dry weight) is a good measurement for the overall efficiency of the symbiosis. Thus, I highly suggest swapping the normalization (per nodule mass for acetylene reduction and biomass per plant) or at least also indicating absolute nodule mass in addition to the existing acetylene reduction assay graph.

I have no doubts that ORS285 is overall the better symbiont for *A. afraspera* than USDA 110 (based on the data presented in Figs 1F, S1, and S2). However, from the currently presented data it is not entirely clear to me, whether USDA 110 is performing poorly in *A. afraspera* or whether ORS285 is the generally more efficient symbiont. The performance of USDA 110 seems very comparable between soybean and *A. afraspera*. It must also be noted here that nodulation kinetics are different for different legumes. 14 dpi for soybean seems a rather early time point and depending on the growth conditions most reports use at least 3-4 weeks old plants. Ideally, the authors could present normalized acetylene reduction data for several time points depending on the nodulation kinetics of the different hosts (e.g. 14 and 21 dpi for soybean and developmentally comparable time points for *A. afraspera*).

L225-232: It was previously shown that *groESL* genes in *B. diazoefficiens* are differentially regulated. Whilst *groESL2* (weak) and 4 (strong) are constitutively expressed, *groESL1* is upregulated upon heat shock and *groESL3* expression is NifA dependent, thus showing higher expression under symbiotic and micro/anaerobic conditions (Fischer HM et al. 1993 EMBO J 127:2901–2912). Do the authors use the same gene nomenclature and if so, they should discuss this found discrepancy of GroESL isoform regulations with the previous study.

Figs 3, 4, and S4: I find the current data presentation of the proteomics and transcriptomics data in heat maps unintuitive and hard to interpret. In its current form, the reader is presented with large heatmaps of. Of the nature of the underlying proteins and genes only rough associations can be made (e.g. upregulated in plants). Only occasionally, the identity of the underlying gene is revealed (Fig 3B) whilst in most cases information is lacking or the indicated examples are not clearly assigned

to a row of the heat map. As such, in most cases I do not know what to make of the individual rows. I suggest the authors find alternative or complementary visualization methods similar to Fig 4C (which should be mentioned in the Figure legend), e.g., Venn diagrams for overall amounts of DAPs and DEGs or Volcano plots if the quantitative differences should be retained. Since such diagrams are likewise only giving a general idea of the overall datasets, additional information of the DEGs and DAPs of each category according to their function might be helpful, e.g. incorporating Panels similar to Fig S4 for each category.

Minor points

The authors occasionally switch from past to present tense when describing the results, please maintain past tense throughout the manuscript.

L42743: "... involving ... membrane permeability ..." reads strange, I suggest changing to "increased membrane permeability" or "membrane permeability changes" or so.

L46: establishes

L57: into the biosphere

L62: Please reconsider the wording of non-natural, which implies a synthetic or artificial interaction. However, the study deals with two wild-type organisms entering a functional (although poorly efficient) symbiosis.

L118: Space between "seemed" and "also"

L119 and Fig S1: How many plants (replicates) were used for the elemental analysis and why are there no error bars on the nitrogen content columns? Did the replicates result in such consistent results that the error bars cannot be drawn?

L137: Please provide a reference for the observation of phosphate accumulation in inefficient nodules.

L272 and Fig 4: Online 272, the authors refer to the blr3675-80 cluster as encoding toluene degradation enzymes, but in Fig 4, the cluster (blr3678-80) is annotated as toluene anabolism.

L275: There seems a word missing about CpxP.

Fig S3: What is the meaning of metabolite xy 1 and xy 2 (e.g. glucose 1 and glucose 2; glutamine 1 and glutamine 2)? Are these different derivatives from GC-MS sample preparation? If so, could they be pooled?

As outlined at the beginning of the comments for the authors, I find this manuscript provides three key messages that add to our understanding of rhizobia-legumes symbioses. Especially the observation that terminal bacteroid differentiation is not necessarily associated with cell enlargement and endoreduplication is an intriguing new facet of this symbiosis. I think the audience of mSphere will be interested by the new aspects of host control over endosymbionts and how bacteria may cope with the concomitant stress, and the global effects that go along with suboptimal host-symbiont pairings. The provided supplemental data further supports the main figures and is crucial in case of tables S1 and S3.

Overall the conclusions made by the authors are supported by the experiments and I may recommend publication, but also think the manuscript can be further strengthened with little effort. My criticism regarding the current manuscripts mainly lays in the data presentation for the omics experiments. I also think the aspect of suboptimal efficiency of USDA 110 with *A. afraspera* compared to either its native host soybean or to the native *A. afraspera* microsymbiont ORS285 could be further clarified by limited time point measurements. Since the method for this is clearly well established in the labs of the authors, I highly suggest the authors perform this straightforward experiment to make sure they compare bacteroids at their peak efficiency.

Dear Editor,

We deeply thank you for your supportive and enthusiastic evaluation of our work and for the decision you made to accept this manuscript in *mSystems* with minor revision. We are very sorry for the long time it took us to prepare the revised version of the manuscript, which was in part due to the current sanitary situation and its consequences on our daily activities.

In the following paragraphs, we provide a point-by-point answer to the reviewers' comments and questions. We hope our answers to their questions, together with the improvements we made in the revised manuscript, will satisfy their expectations.

Response to Reviewer 1:

We would like to thank Reviewer 1 for his/her analysis of our work and his/her advices to improve the quality of the manuscript.

Major points :

*L116 and Fig 1D, E: Reviewer 1 questioned the presentation of the data regarding the analysis of symbiotic efficiency in the selected symbiotic systems supporting the idea that USDA110 is a poor symbiotic partner for *Aeschynomene afрасpera*. Our way to normalize the gain of plant biomass attributable to the symbiotic process allows the comparison of plants that have intrinsically very different size/mass (Fig. 1D). Indeed, in this graph we compare soybean and *Aeschynomene*. Regarding the normalization of the ARA (Fig. 1E), we unfortunately did not measure plant biomass in the same experiment. Thus, we apologize that normalizing the ARA on nodule biomass is not possible with the data we have in hands. However, we added a picture of *A. afрасpera* plants at later time points (21 dpi), where the difference in symbiotic efficiency of the two strains is more pronounced (Fig. 1C), strengthening our conclusion that the symbiosis between USDA110 and *A. afрасpera* is suboptimal.*

*Reviewer 1 points out that we work on early time points of the symbiotic process in the selected symbiotic systems. However, it has been shown that nodules fix nitrogen as of 5 dpi in *Aeschynomene afрасpera* (Bonaldi et al., 2011, MPMI) and that soybean nodules infected with USDA110 reach 95% of their nitrogen fixation activity as of 13 dpi (Pessi et al., 2007, MPMI). Thus, we focused our analysis on 14dpi plants, both at the phenotypical/physiological and molecular levels.*

L225-232: We'd like to thank Reviewer 1 for spotting that we made a mistake during the genome reannotation procedure, as we introduced a step of automatic numbering of the genes belonging to multigene families, which ended up in a gene nomenclature that was not consistent with previously published data in USDA110. So, we modified our Supplementary Table 1 and removed this automatic numbering to restore the labelling that was previously accepted in the community (we added columns in the table that include the different genome annotations and their correspondence). So, indeed, the GroESL proteins that are over-accumulated during symbiosis are the very same that Prof. Hans-Martin Fischer and colleagues saw upregulated in 1993 in soybean. We apologize for this mistake and we made all necessary corrections throughout the manuscript, figures and tables.

Figs 3, 4, and S4: Reviewer 1 questioned the way we present the omics data, which is always a puzzling question, especially when multiple datasets are presented within a single paper. For the presentation of the omics data in figures 3, 4 and S4, we chose heatmaps that provide a global view of the data. Indeed, we tried to make it simple and avoided to add too many information on the graphs. For a more detailed analysis of the given gene sets, the supplementary tables 1 and 3 can be used. To make this easier, we added a column called "Fig 3 panel" in Table S1 to specify which gene appears in Figure 3 as "induced_in_planta", as "induced_in_AA" or as "induced_in_GM". Accordingly, we added a column in Table S1 specifying the COG class of each gene.

Minor points:

L62: we removed "non natural" and rephrased as follows : We combined multi-omics with physiological analyses to show that the non-natural symbiotic couple formed by *Bradyrhizobium diazoefficiens* USDA110 and *Aeschynomene afraspera*, in which host and symbiont did not evolve together, is functional but displays a low symbiotic efficiency associated to a disconnection of terminal bacteroid differentiation features.

L119 and Fig S1: The experiment was performed by pooling 30 plants per condition, so we did not obtain biological replicates for this experiment but a single value for the pool of plants. As it was consistent with the other physiological data, we included this single experiment as supplemental figure.

L137: a reference was added (Lamouche et al., 2019 – ref 18).

Fig S3: Reviewer 1 asked why are there glucose 1 and 2 and glutamine 1 and 2. These different fragments of the same molecule are detected by the mass spectrometer. They vary in similar ways over the different conditions, so we could indeed have shown only one per metabolite. We thought that presenting both shows the consistency of the metabolomic analysis.

We took in account all remarks regarding wording (L42-43, L46, L57, L118, L275), past/present tense, typos, and we thank Reviewer 1 for spotting them.

Response to Reviewer 2:

We thank Reviewer 2 for his/her evaluation of our work and for sharing constructive comments.

In addition to the points spotted by the reviewers, we made all the datasets available online, and the links are provided in a specific paragraph at the end of the manuscript.

In conclusion, we hope that this new version of the manuscript with all the improvements it contains will be accepted for publication in mSystems.

Best regards,

Benoit Alunni

April 14, 2021

Dr. Benoit Alunni
Université Paris-Saclay
I2BC - CEA-CNRS-Université Paris Sud
Bat 34. Avenue de la Terrasse
Gif-sur-Yvette 91198
France

Re: mSystems01237-20R1 (*Bradyrhizobium diazoefficiens* USDA110 nodulation of *Aeschynomene afraspera* is associated with atypical terminal bacteroid differentiation and suboptimal symbiotic efficiency)

Dear Dr. Benoit Alunni:

I am sorry for the delay in sending the decision. It has been a struggle for me to keep up with my editorial duties due to my children being home for remote schooling. Thank you for your patience and congratulations on your paper's acceptance.

Your manuscript has been accepted, and I am forwarding it to the ASM Journals Department for publication. For your reference, ASM Journals' address is given below. Before it can be scheduled for publication, your manuscript will be checked by the mSystems senior production editor, Ellie Ghatineh, to make sure that all elements meet the technical requirements for publication. She will contact you if anything needs to be revised before copyediting and production can begin. Otherwise, you will be notified when your proofs are ready to be viewed.

- Minimum resolution of 1280 x 720
- .mov or .mp4. video format
- Provide video in the highest quality possible, but do not exceed 1080p

· Provide a still/profile picture that is 640 (w) x 720 (h) max

We recognize that the video files can become quite large, and so to avoid quality loss ASM suggests sending the video file via <https://www.wetransfer.com/>. When you have a final version of the video and the still ready to share, please send it to Ellie Ghatineh at eghatineh@asmusa.org.

Sincerely,

Michelle Heck
Editor, mSystems

Journals Department
Figure S2: Accept
Figure S7: Accept
Table S3: Accept
Figure S1: Accept
Figure S6: Accept
Table S1: Accept
Figure S5: Accept
Table S2: Accept
Figure S4: Accept
Figure S3: Accept